# Single-Facility Analysis of COVID-19 Status of Healthcare Employees during the Eighth and Ninth Pandemic Waves in Japan after Introducing Regular Rapid Antigen Testing

**DOI:** 10.3390/vaccines12060645

**Published:** 2024-06-09

**Authors:** Masayuki Nagasawa, Tomoyuki Kato, Hayato Sakaguchi, Ippei Tanaka, Mami Watanabe, Yoko Hiroshima, Mie Sakurai

**Affiliations:** 1Department of Infection Control, Musashino Red Cross Hospital, 1-26-1, Kyonan-cho, Musashino 180-8610, Tokyo, Japan; ict-ph2@musashino.jrc.or.jp (T.K.); h.sakaguti@musashino.jrc.or.jp (H.S.); i.tanaka@musashino.jrc.or.jp (I.T.); mami36@musashino.jrc.or.jp (M.W.); y.hiroshima@musashino.jrc.or.jp (Y.H.); mie1214@musashino.jrc.or.jp (M.S.); 2Department of Pediatrics, Musashino Red Cross Hospital, 1-26-1, Kyonan-cho, Musashino 180-8610, Tokyo, Japan; 3Department of Pharmacy, Musashino Red Cross Hospital, 1-26-1, Kyonan-cho, Musashino 180-8610, Tokyo, Japan; 4Department of Laboratory, Musashino Red Cross Hospital, 1-26-1, Kyonan-cho, Musashino 180-8610, Tokyo, Japan; 5Department of Nursery, Musashino Red Cross Hospital, 1-26-1, Kyonan-cho, Musashino 180-8610, Tokyo, Japan

**Keywords:** COVID-19, Omicron strain, rapid antigen test, healthcare employee, notification system

## Abstract

Background: Community infections of severe acute respiratory syndrome coronavirus 2 (SARS-CoV-2) have increased rapidly since the emergence of the Omicron strain. During the eighth and ninth pandemic waves—when movement restrictions in the community were eased—the all-case registration system was changed, and the actual status of infection became uncertain. Methods: We conducted regular rapid antigen tests (R-RATs) once or twice a week as self-testing to examine the actual state of coronavirus disease (COVID-19) diagnosis among healthcare employees. Results: Overall, 320 (1.42/day) and 299 (1.76/day) employees were infected in the eighth and ninth pandemic waves. During both periods, 59/263 doctors (22.4%), 335/806 nurses (41.6%), 92/194 administrative employees (47.4%), and 129/218 clinical laboratory technicians (59.2%) were infected. In the eighth wave, 56 of 195 employees were infected through close contact; in the ninth wave, 26 of 62 employees were infected. No significant difference was observed in the number of vaccinations between infected and non-infected employees. The positivity rate of R-RATs was 0.41% and 0.45% in the eighth and ninth waves. R-RATs detected infection in 212 and 229 employees during the eighth and ninth waves, respectively; the ratio of R-RAT-detected positive employees to those who reported infection was significantly higher during the ninth wave (odds ratio: 1.67, 95% confidence interval: 1.17–2.37, *p* < 0.001). Conclusions: The number of infected healthcare employees remained high during the eighth and ninth pandemic waves in Japan. The R-RAT is considered effective for detecting mild or asymptomatic COVID-19 at an early stage and at a high rate in healthcare employees.

## 1. Introduction

Since early 2020, coronavirus disease (COVID-19) has become a global crisis of the 21st century, causing a major pandemic that affects human health and activities worldwide and leading to a major international emergency [1,2]. In December 2022, the number of severe acute respiratory syndrome coronavirus 2 (SARS-CoV-2)-infected people worldwide exceeded 630 million, with the death toll exceeding approximately 6.5 million. In Japan, the cumulative number of infected people exceeded 30 million, and the death toll exceeded 70,000 as of May 2023 [3]. Notably, the COVID-19 pandemic has severely affected skilled nursing facility residents and healthcare workers [4,5]. Healthcare workers are at a high risk of SARS-CoV-2 exposure during patient care [6,7] and are among the earliest groups prioritized for COVID-19 vaccination [8].

The emergence of the Omicron strain, which has stronger infectivity, resulted in an outbreak in the sixth pandemic wave; the seventh wave became the largest outbreak to date, causing the highest number of deaths in Japan [3]. Several clusters also occurred in medical institutions. The impact on general medical care was serious, with medical workers being infected or coming into close contact with the virus. Considering the impact of COVID-19 outbreak within medical institutions, stricter infection control measures than those for the general public are mandatory.

On the other hand, owing to widespread vaccine use and weakening of the toxicity of the virus in healthy people, behavioral restrictions and infection control measures in the general public are being relaxed, and society as a whole is returning to the pre-COVID-19 pandemic situation. Simultaneously, the system for reporting all cases of infection in Japan changed, making it difficult to understand the actual COVID-19 status in the community.

Under these circumstances, our hospital introduced a regular rapid antigen test (R-RAT) for healthcare workers as self-testing during the eighth and ninth pandemic waves.

## 2. Methods

The Musashino Red Cross Hospital is a tertiary emergency medical facility in the North Tama area of Tokyo, Japan. It has 611 beds, more than 20,000 annual admissions, more than 11,000 annual emergency transfers, and approximately 1800 outpatients per day. In our hospital, 263 doctors, 806 nurses, 194 administrative employees, and 218 clinical laboratory technicians work as regular employees.

### 2.1. Notification System

To understand the health status of employees, prevent nosocomial infections, and conduct epidemiological investigations, when an employee is absent from work because of an infectious disease, a report must be submitted to the Infection Control Office by the supervisor as an infection control measure in our hospital. Additionally, since the beginning of the COVID-19 pandemic, we decided to notify anyone who has come into close contact with COVID-19 patients and required them to undergo follow-up observation for a designated period.

We referred to the Japanese Ministry of Health, Labor, and Welfare and defined “close contact” as “an infected person within the same household” or “contact for more than 15 min, such as having a meal with an infected person or conversation without wearing a mask [9]”. In such cases, a close contact report must be submitted. Until the seventh wave, the department managers used paper-based notification methods. The Infection Control Office confirmed the report details and made a final decision, provided instructions, and notified the employee when they could return to resume work. Since the sixth wave, the number of infected employees increased rapidly; therefore, we started the digitization of notifications using the hospital intra-net system to quickly respond and give instructions to close contacts and infected employees. This study primarily analyzed the eighth and ninth waves after switching to an electronic reporting system.

The reported contents were as follows: department, age, gender, date of onset, symptoms, presence or absence of meals outside with others, presence of family members with symptoms/infection, behavioral history up to 2 days before the onset of symptoms, and free description.

### 2.2. Regular Antigen Qualitative Test

From 21 August 2022, with the end of the seventh wave, we started the R-RATs (once to twice/week), using rapid COVID-19 antigen test kits provided by the Tokyo Metropolitan Government. The RAT kits were distributed to the employees, who tested themselves once or twice a week as self-testing and reported the results to their supervisor. Subsequently, the supervisor reported the weekly results to the office. The testing dates were not specified and were left to each employee. We used the following three RAT kits approved by the Japanese Ministry of Health, Labor, and Welfare: (1) Panbio COVID-19 Antigen Rapid Test (Abott Japan LLC, Tokyo, Japan); (2) Rapid COVID-19 Antigen Self-Test (Siemens Healthinners Diagnostics, Tokyo, Japan); and (3) SARS-CoV-2 Rapid Antigen Test (Roche Diagnostics, Tokyo, Japan). Based on the data provided by the distributors, the sensitivities and specificities of these RAT kits were almost identical.

### 2.3. Criteria for Release from Quarantine

During the eighth and ninth waves, “close contact” employees could work while confirming a negative RAT every day until the third day. “Infected employees” could return to work if a negative RAT result was confirmed after the fifth day with no fever for more than 24 h. Otherwise, infected employees could return to work after the 10th day, even if they did not confirm negative RAT results. For asymptomatic infected employees, the date of symptom onset was defined as the date of a positive test result.

We used the average number of employees during the analysis period, including 263 doctors, 806 nurses, 194 administrative employees, and 218 clinical laboratory technicians.

### 2.4. Statistical Analysis

Chi square and Mann–Whitney analyses were used for statistical analysis, and a *p*-value < 0.05 was considered significant. Statistical analyses were performed using JMP 14 software (SAS Institute, Cary, NC, USA).

## 3. Results

### 3.1. Analysis from the Infectious Disease Notification Reporting System

Table 1 shows trends in the number of new SARS-CoV-2 infections among employees and patients in the southern North Tama Medical Area, where our hospital operates as a core and designated type 2 infectious disease hospital, since the third wave of the pandemic. After the Omicron strain’s emergence during the sixth wave, the number of infected employees rapidly increased. As the R-RAT for employees was initiated on 21 August 2022, we compared data obtained during the eighth (20 September 2022–2 May 2023; 225 days) and the ninth waves (7 May 2023–21 October 2023; 168 days). The number of reported infections in the eighth and ninth waves was comparable, at 320 and 299, respectively (Table 1), with no significant difference in the number of reported cases per day (1.42 vs. 1.76, odds ratio [OR] 1.25, 95% confidence interval [CI]: 0.97–1.61, *p* = 0.08). During the COVID-19 pandemic, the total population of the southern North Tama Medical Area was approximately 1.06 million people, and the average number of employees at our hospital was 1481 individuals, including 263 doctors, 806 nurses, 194 administrative employees, and 218 clinical laboratory technicians; if the infection rate was the same in both population, the ratio was approximately 1.39 × 10^−3^. Therefore, the infection rate of our employees was higher than that in the general population during the third wave; lower during the fourth and fifth waves; and comparable during the sixth and seventh waves, after the appearance of the Omicron strain. In the eighth wave, the proportion of infected employees increased significantly compared with that of the general population. However, in the eighth wave, the previous all-case registration reporting system was changed to a voluntary online registration system in which individuals could register online using commercially available RAT without visiting a medical institution. Since the new system allowed for voluntary registration, a significant number of asymptomatic and mildly infected people were assumed to not have registered. In the ninth wave, the reporting system was changed to fixed-point reporting, making conducting comparative studies, such as the one above, impossible.

Table 2 presents the initial symptoms of the infected employees during the eighth and ninth waves. During both waves, fever and sore throat were the most common symptoms (60–70%), followed by cough and headache. Several symptoms were milder than those observed during previous epidemics. Asymptomatic cases accounted for 7.8% and 5.7% in the eighth and ninth waves, respectively. In cases of infection through close contact, the proportion of asymptomatic cases was high, at 33.9% and 15.4%, respectively; the rate was significantly higher than that of infected employees without close contact; in the eighth wave (33.9% [19/56] vs. 2.3% [6/264]) and in the ninth wave (15.4% [4/26] vs. 4.8% [13/273]). Our regulation may have effectively detected the number of asymptomatic positive cases.

Among the infected employees, nurses accounted for the majority (Table 3). The trend was comparable between the eighth and ninth waves. Regarding the infection rate by occupation (infected employees vs. total employees), the infection rate was significantly lower in doctors than in other occupations.

During the eighth wave, 195 people had close contact with SARS-CoV-2; among them, 56 were infected. Durin the ninth wave, 36 of 62 close contacts developed infections (Table 2). The transmission rate from close contacts was lower in the eighth wave than in the ninth wave, although no significant difference was observed (0.29 vs. 0.42, *p* = 0.052). Regarding the ratio of close contacts to total infected employees, the number of close contacts was significantly higher in the eighth wave (odds ratio: 2.94, 95%CI: 2.12–4.07, *p* < 0.0001) than in the ninth wave. Upon comparing the number of vaccinations received by infected and non-infected close contacts, no significant difference was observed during each wave (eighth wave: 3.41 vs. 3.56, ninth wave: 3.88 vs. 4.42).

Nevertheless, of the 299 employees infected during the ninth wave, 47 had a history of infection. Subsequently, we compared the number of vaccinations administered to employees with and without a history of infection. We found that 40 of 283 employees vaccinated with ≥3 doses and 7 of 15 employees vaccinated with <3 doses had a history of infection; thus, those vaccinated with ≥3 doses were less frequently infected (OR: 0.30, 95% CI: 0.12–0.79, *p* < 0.0001). In the eighth wave, only 15/320 infected employees had a history of COVID-19.

Regarding the infection route, family infection was the most common (78 employees [24.4%] in the eighth wave and 55 employees [18.4%] in the ninth wave), followed by outside meals (32 employees [10.0%] and 54 employees [18.1%] in the eighth and ninth waves, respectively).

The number of infected employees for whom we were able to examine the trends in the number of days until antigen negativity was 180 and 181 in the eighth and ninth waves, respectively. Compared with those in the eighth wave (Figure 1), infected medical employees became negative for RAT a little less than a day earlier in the ninth wave, showing that COVID-19 became even more attenuated from the eighth to the ninth wave.

### 3.2. Analysis of Regular Rapid Antigen Testing

Figure 2 shows trends in the number of positive cases and positivity rates of R-RAT for employees and those in the number of test reports by occupation. At the peak of the pandemic, the positivity rate exceeded 1.00%, and the number of positive cases exceeded 30 per week. Doctors were tested less than once/week (0.84 ± 0.35/week), whereas individuals with other occupations were tested 1.5–2 times/week (nurses: 1.41 ± 0.37/week, administrative employees: 1.72 ± 0.30/week, and clinical laboratory technicians and others: 1.64 ± 0.59/week). Notably, the ratio of the number of positive cases on R-RAT to the number of reported employees with infection was lower in doctors, who underwent less frequent R-RAT than other occupational groups.

## 4. Discussion

The digitization of employee infection reports in our hospital resulted in a uniform, qualified, simplified, and faster reporting of the contents, making the reports more detailed and easier to analyze. Therefore, we examined and analyzed the actual COVID-19 status among hospital employees during the COVID-19 pandemic, focusing on the eighth and ninth waves, during which the number of infected employees rapidly increased. To prevent infection within hospitals, universal masking is mandatory inside hospitals, and conversations and meals between employees are prohibited without masks, and this regulation was not eased until the ninth wave.

Until the third wave, all employees diagnosed with COVID-19 at our hospital were nurses and doctors working in the COVID-19 wards; thus, these cases were a nosocomial infection occurring because of medical activities. Community transmission of COVID-19 was detected since the fifth wave and expanded widely with the emergence of the Omicron strain during the sixth wave. Consequently, the number of infections among employees, including administrative workers, increased. Immediately before the fifth wave, mRNA vaccines for COVID-19 were first launched for individuals aged ≥ 65 years and healthcare workers. Vaccine effectiveness may have been one reason why the number of infected people aged ≥ 65 years during the fifth wave remained low in Japan [10], and the number of infected employees at our hospital also remained low. Nevertheless, nationwide statistics reported that the number of severe cases and death rates among relatively healthy infected people aged 40–65 years, who were not eligible for vaccination, were the highest during the fifth wave [10].

Since the sixth wave, when community infections increased rapidly owing to the Omicron strain [11], which had increased infectivity due to viral mutations and stronger ability to evade immunity from vaccines [12,13,14], the chances of community and home infections were speculated to be increased among medical workers and hospital employees, and the probability of infection became almost the same as that observed in the community (Table 1). The lower infection rate among doctors than that for other occupations may be due to the thorough infection prevention measures taken when examining patients and in daily life. On the other hand, the high infection rate among nurses may be because they were required to have closer and longer contact with infected patients than doctors, making their infection risk higher. The suppression of infection rates in nurses to the same level as that in the general community or lower than that of administrative workers and clinical laboratory technicians in the same hospital could be attributed to the effectiveness of infection prevention measures among nurses.

After the sixth wave, the pneumonia symptoms of COVID-19 tended to be milder than those before [15,16,17,18,19]. Since the eighth wave, the number of beds for critically ill patients decreased, indicating that fewer COVID-19 patients among healthy individuals required hospitalization. Nevertheless, the number of patients requiring long-term hospitalization owing to worsening of pre-existing complications increased during this period. For the general public, the vaccine may be able to reduce the severity of COVID-19 symptoms; however, it is not sufficient to prevent the infection. Upon comparison of close contacts (Table 2), we found no significant difference in the number of vaccinations between infected and non-infected individuals, indicating that vaccination was ineffective in preventing infections due to intense exposure. Similar results were reported in in a previous report [20]. However, during the ninth wave, employees who had been fully vaccinated (≥3 times) had significantly fewer number of previous infections, indicating that vaccination had a certain preventive effect.

Infections caused by the Omicron strain and its derivatives, which have acquired mutations allowing them to evade vaccines [12,13,14], do not cause serious illness in the general public but may cause severe infection in older people with comorbidities and those with severe immunodeficiency [21]. After the sixth wave, infection control measures at medical institutions and nursing homes treating this specific population have become increasingly difficult, and mass infections in elderly welfare facilities and welfare facilities for people with disabilities have increased rapidly [10]. However, when the reported mortality rate due to COVID-19 among older people during the seventh wave was equal to or lower than that of seasonal influenza, the relaxation of behavior and infection control measures in general social life was recommended. Until then, the national registration system was changed to allow voluntary personal online registration for people who were diagnosed at home using a commercially available RAT kit, although patients diagnosed at medical institutions remained registered by doctors. Owing to the lowered sense of crisis in the society and relaxation of social behavioral norms, there would have been an increase in cases where asymptomatic or mild symptoms were not tested or diagnosed, causing concerns that the actual situation could not be ascertained. This may be related to our finding that the ratio of infection rate among hospital employees to the infection rate in the community increased by approximately three times in the eighth wave (Table 1). With the relaxation of infection control measures, R-RAT was started with cooperation from the Tokyo Metropolitan Government. We found that some employees were completely asymptomatic (Table 2). When employees come into close contact, the detection rate of asymptomatic infected people increased if the RAT was performed daily during the incubation period. Our results suggested that there must have been a considerable number of asymptomatic infected individuals in the community. Additionally, the infection rate among close contacts increased from 28.7% (56/195) in the eighth wave to 41.9% (26/62) in the ninth wave, indicating that the virus became even more infectious during the ninth wave. The ratio of close contacts to infected people in the eighth wave (195 vs. 320) was significantly higher than in that the ninth wave (62 vs. 299) (odds ratio: 2.94, 95%CI: 2.12–4.07, *p* < 0.0001). This may be because in the ninth wave, there were many asymptomatic infected people, and close contacts in several cases were not recognized. As shown in Table 3 and Figure 2, R-RAT twice/week had a better infection detection rate than R-RAT once/week. Asymptomatic infected individuals without close contact accounted for a small percentage of all infected employees (Table 2). However, several initial symptoms were relatively mild and temporary, and many infected cases might have been overlooked without the R-RAT. During the eighth and ninth waves, when infectivity increased and clinical symptoms became milder, the R-RAT for employees was considered to have had a certain effect from the perspective of preventing nosocomial infections. Notably, during the ninth wave, the proportion of infected employees identified through the R-RAT increased compared with that during the eighth wave in all occupations.

Although the risk to healthy people has decreased to the same level as that of seasonal influenza, COVID-19 remains an infectious disease that requires caution in medical institutions where older people with complications are hospitalized. Therefore, understanding the actual state of infection among medical workers involved in patient treatment and care is important. This observational study showed that regular testing using the RAT kit was effective in detecting infections among healthcare workers.

This study had some limitations. First, this was a retrospective observational study. Second, the implementation of the R-RAT is left to each medical employee, making it impossible to confirm whether each person has been tested evenly and equally. The notification system was based on interviews with the head of each department, and there may have been variations in the content and accuracy. Finally, the timing of confirmation of a negative RAT is not mandatory and is based on each employee’s discretion.

## 5. Conclusions

The R-RAT effectively detects mild or asymptomatic COVID-19 at an early stage and at a high rate in healthcare employees during the eighth and ninth pandemic waves in Japan.

## Figures and Tables

**Figure 1 vaccines-12-00645-f001:**
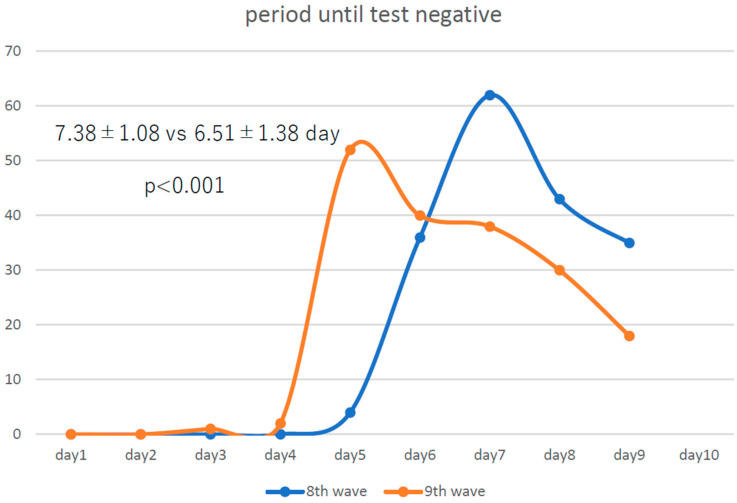
“Infected employees” were requested to present a negative rapid antigen test (RAT) result after the fifth day with no fever for more than 24 h to return to work. A total of 180 and 181 employees during the eighth and ninth waves, respectively, were eligible for this analysis. The period until a negative RAT in the ninth wave was significantly shorter than that in the eighth wave.

**Figure 2 vaccines-12-00645-f002:**
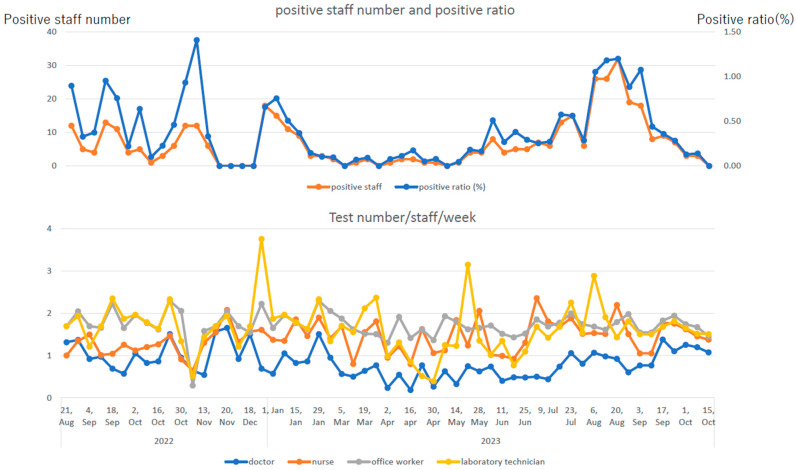
Trends in the number of positive cases and positivity rate of regular rapid antigen test (R-RAT) results for employees (**upper graph**) and trends in the number of test reports by occupation (**lower graph**). The number of positive employees and positivity rate correlate well with the epidemic situation in the medical area . The number of tests for doctors is lower than that for other professions, at approximately half.

**Table 1 vaccines-12-00645-t001:** Comparison of trends in the number of infected employees and number of infected individuals in the medical area from the third to the ninth pandemic wave.

	Start	End	Period (Days)	Infected Staff/Day	Infected Staff	Registered Patients in Southern North- Tama Area	Infected Staff/Registered Patients × 1000
3rd wave	20 November 2020	19 March 2021	120	0.14	17	4500	3.78
4th wave	20 March 2021	30 June 2021	103	0.01	1	2939	0.34
5th wave	3 July 2021	25 September 2021	85	0.12	10	10,434	0.96
6th wave	1 January 2022	30 May 2022	150	0.65	98	60,061	1.63
7th wave	25 June 2022	19 September 2022	87	1.46	127	84,532	1.50
8th wave	20 September 2022	2 May 2023	225	1.42	320	68,744 *	4.65 *
9th wave	8 May 2023	24 October 2023	170	1.76	299	n.a.	n.a.

*: Registration method changed from full registration to partial application registration; n.a.: not applicable.

**Table 2 vaccines-12-00645-t002:** Clinical symptoms of infected employees and those infected through close contact during the eighth and ninth pandemic waves.

		Total	Fever	Sore Throat	Cough	Headache	Nothing
8th wave	Infected staff	320	21065.6%	21366.6%	12137.8%	5216.3%	257.8%
Close contact	195					
	Close contact 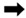 Infected	56	25	28	18	9	19
			44.6%	50.0%	32.1%	16.1%	33.9%
9th wave	Infected staff	299	21571.9%	17558.5%	10535.1%	5618.7%	175.7%
Close contact	62					
	Close contact 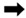 Infected	26	13	10	5	2	4
			50.0%	38.4%	35.1%	7.7%	15.4%

**Table 3 vaccines-12-00645-t003:** Number of infected employees registered and that detected through regular rapid antigen tests by occupations during the eighth and ninth pandemic waves.

8th Wave	Infected Staff Registered	Ratio (%) *	Odds Ratio (vs. Doctor)	Positive Staff in Regular Ag Test	Ratio **
Doctor	35	13.3	1.00	16	0.46
Nurse	168	20.8	1.57 ^#^	114	0.68
Administrative worker	44	22.7	1.70 ^#^	27	0.61
Laboratory technician	73	33.4	2.52 ^$^	55	0.75
**9th Wave**	**Infected Staff Registered**	**Ratio (%) ***	**Odds Ratio** **(vs. Doctor)**	**Positive Staff in Regular Ag Test**	**Ratio ****
Doctor	24	4.3	1.00	16	0.67
Nurse	167	20.7	2.27 ^$^	130	0.78
Administrative worker	48	24.7	2.71 ^$^	35	0.73
Laboratory technician	56	25.7	2.81 ^$^	48	0.86

*: infected staff registered/total staff; #: *p* < 0.05, $: *p* < 0.001 (chi square test); **: positive staff in regular Ag test/infected staff registered.

## Data Availability

Data available on request from the authors.

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
