# Peer review of "Single-Facility Analysis of COVID-19 Status of Healthcare Employees during the Eighth and Ninth Pandemic Waves in Japan after Introducing Regular Rapid Antigen Testing"

_vaccines, 2024, doi:10.3390/vaccines12060645_

Round 1
Reviewer 1 Report
Comments and Suggestions for Authors
This is a straightforward, carefully reported retrospective study of the experience of multiple waves of COVID-19 and the effectiveness of R-RAT in detecting changes in the infectivity of different waves. No changes needed except to conclude with clear advice to health authorities how and when to control activities that lead to infection especially in high risk medical personnel.
Author Response
Dear Reviewer 1
We would like to thank you for carefully reading our paper and for your valuable advice.
I corrected the points you commented out as follows:
Revised parts are highlighted in blue in the manuscript for easy identification.
Reviewer 1:
Comments and Suggestions for Authors
This is a straightforward, carefully reported retrospective study of the experience of multiple waves of COVID-19 and the effectiveness of R-RAT in detecting changes in the infectivity of different waves. No changes needed except to conclude with clear advice to health authorities how and when to control activities that lead to infection especially in high risk medical personnel.
The difference in the weight of community-acquired infections and nosocomial infections is not limited to COVID-19. Medical professionals are always required to take thorough measures to prevent infection, but this is not an easy task. I hope that this research will be useful in the future when thinking about infection control measures for medical workers, but I think it would be excessive to include it as a message in the conclusion, so I will omit it.
We appreciate your review of our responses and revised versions. We sincerely hope that the revised version is satisfactory to the reviewers and deserves to be accepted.
Sincerely,
Corresponding author:
Authors
Reviewer 2 Report
Comments and Suggestions for Authors
Dear Editor,
Thank you for the opportunity to review the manuscript entitled "Single-facility analysis of COVID-19 status of healthcare employee during the eighth and ninth pandemic waves in Japan after introducing regular rapid antigen testing".
The article deals with the implemented measures for control of COVID-19 in hospital settings in a single hospital in Japan. Although data on self-testing with RAT from only one hospital are presented, these data are of great importance. The manuscript is interesting and clearly presented. There are some limitations, and the authors have addressed them in the Discussion. However, there are only few issues, and the following comments describe them.
- Line 66: "our hospital introduced a regular rapid antigen test (R-RAT) for healthcare workers..." Please consider describing that R-RAT was performed as self-testing.
- Line 95: again you may consider to more clearly explain that R-Rat testing was performed as self-testing.
- And finally, authors may also consider mentioning self-testing in the abstract as well.
These revisions would enhance the clarity and accuracy of the manuscript.
Best regards!
Author Response
Dear Reviewer 2
We would like to thank you for carefully reading our paper and for your valuable advice.
I corrected the points you commented out as follows:
Revised parts are highlighted in blue in the manuscript for easy identification.
Reviewer 2:
Comments and Suggestions for Authors
Dear Editor,
Thank you for the opportunity to review the manuscript entitled "Single-facility analysis of COVID-19 status of healthcare employee during the eighth and ninth pandemic waves in Japan after introducing regular rapid antigen testing".
The article deals with the implemented measures for control of COVID-19 in hospital settings in a single hospital in Japan. Although data on self-testing with RAT from only one hospital are presented, these data are of great importance. The manuscript is interesting and clearly presented. There are some limitations, and the authors have addressed them in the Discussion. However, there are only few issues, and the following comments describe them.
Line 66: "our hospital introduced a regular rapid antigen test (R-RAT) for healthcare workers..." Please consider describing that R-RAT was performed as self-testing.
Line 95: again you may consider to more clearly explain that R-Rat testing was performed as self-testing.
And finally, authors may also consider mentioning self-testing in the abstract as well.
These revisions would enhance the clarity and accuracy of the manuscript.
I added the “self-testing” comment in each designated part. (line 23, line 68, and line 102)
Best regards!
We appreciate your review of our responses and revised versions. We sincerely hope that the revised version is satisfactory to the reviewers and deserves to be accepted.
Sincerely,
Corresponding author:
Authors
Reviewer 3 Report
Comments and Suggestions for Authors
The authors present a retrospective study of infections among hospital workers during several pandemics of COVID 19 in Japan. The data presented in Table 1 indicates that employee infection was higher than the general population during the third wave, lower in fourth and fifth waves and higher in eighth and ninth waves. Authors, were the infection control protocols implemented by the hospital the same for all 9 waves or did it vary? Curious as to whether earlier control protocols were not as rigid due to the disease being new?
Do you attribute the higher rates of infection in the eighth and ninths waves to the increased transmissibility of the Omicron variant? It is unfortunate that asymptomatic and/or mildly infected people probably did not report their infection and probably did not due the RAT test to determine? This skews the data somewhat but what can you do, you have to use the retrospective data available.
Title to Table 3 needs attention "Number of infected employees and that infected..." Not sure if there is a word missing between that and infected?
Lines 170-178, how did you verify that the infected individuals had close contact with and infected individual?
Lines 186-189, does this mean that most of your hospital employees were infected through contact outside the hospital?
Data in Figure 3 indicates doctors with the lowest trend in positive cases probably because they had less prolonged contact with patients compared to nurses. Laboratory technicians handle the fluid samples so can be exposed multiple times to the virus. Why is the incidence among office workers higher or similar to those of the nurses, this seems unusual in that office workers would be most isolated from infected patients?
Overall a good paper with information of value for hospital staffs related to the incidence of COVID 19 among various levels of employees. Data supports conclusions and discussion is informative.
Author Response
Dear Reviewer 3
We would like to thank you for carefully reading our paper and for your valuable advice.
I corrected the points you commented out as follows:
Revised parts are highlighted in blue in the manuscript for easy identification.
Reviewer 3:
Comments and Suggestions for Authors
The authors present a retrospective study of infections among hospital workers during several pandemics of COVID 19 in Japan. The data presented in Table 1 indicates that employee infection was higher than the general population during the third wave, lower in fourth and fifth waves and higher in eighth and ninth waves. Authors, were the infection control protocols implemented by the hospital the same for all 9 waves or did it vary? Curious as to whether earlier control protocols were not as rigid due to the disease being new?
Infection control protocol implemented at hospitals have been consistently thorough since the first wave, and were not relaxed during the eighth and ninth waves. Inside the hospital, universal masking was strictly enforced and silent eating was strictly enforced. Furthermore, since December 2020, just before the onset of 3rd wave of the pandemic, highly sensitive antigen quantitative tests have been introduced, and a rapid testing system for symptomatic employees has been established, making it possible to quickly catch infected employees. (line 227-230)
Do you attribute the higher rates of infection in the eighth and ninths waves to the increased transmissibility of the Omicron variant? It is unfortunate that asymptomatic and/or mildly infected people probably did not report their infection and probably did not due the RAT test to determine? This skews the data somewhat but what can you do, you have to use the retrospective data available.
As discussed in the paper, we think that the increase in the frequency of infections among healthcare workers during the 8th and 9th waves is due to the increased infectivity of the Omicron strain and the increase in community-acquired infections, which increased opportunities for healthcare workers to become infected in the community. (line 244-249)
Title to Table 3 needs attention "Number of infected employees and that infected..." Not sure if there is a word missing between that and infected?
I added an appropriate word to make it sure to understand.
Number of infected employees and that detected through regular rapid antigen tests by occupations during the eighth and ninth pandemic waves.
âž¡
Number of infected employees registered and that detected through regular rapid antigen tests by occupations during the eighth and ninth pandemic waves. (line 171)
Lines 170-178, how did you verify that the infected individuals had close contact with and infected individual?
In our infectious disease notification system, healthcare employees are required to report cases that are judged to have been a close contact, even if the person has not developed symptoms. (line 86)
Lines 186-189, does this mean that most of your hospital employees were infected through contact outside the hospital?
Determining the route of infection is extremely difficult. In particular, we believe that it is next to impossible to prove infection in the community. We believe that the increase in cases with unknown infection routes indirectly indicates that community-based infections have increased. However, this does not rule out infection within the hospital. (line 244-249, line 292-299)
Data in Figure 3 indicates doctors with the lowest trend in positive cases probably because they had less prolonged contact with patients compared to nurses. Laboratory technicians handle the fluid samples so can be exposed multiple times to the virus. Why is the incidence among office workers higher or similar to those of the nurses, this seems unusual in that office workers would be most isolated from infected patients?
I agree with your comment. It is understandable that nurses, who often have close and long-term contact with patients, have a higher infection rate than doctors.
We believe that the high number of infections among office workers, who have few opportunities to come into contact with patients, indirectly indicates that the number of infections in the community has increased.  Based on the above points, we believe that nurses and other healthcare workers are more careful about infection control measures than the general public, even in their general social life. (line 249-256)
Overall a good paper with information of value for hospital staffs related to the incidence of COVID 19 among various levels of employees. Data supports conclusions and discussion is informative.
We appreciate your review of our responses and revised versions. We sincerely hope that the revised version is satisfactory to the reviewers and deserves to be accepted.
Sincerely,
Corresponding author:
Authors
Reviewer 4 Report
Comments and Suggestions for Authors
Estimated Authors,
I've read with great interest your paper reporting on a single center experience during the later waves of COVID-19 pandemic in Japan. The present paper stresses how later waves, associated with Omicron VOC, were less frequently acknowledged for their morbidity and mortality, even though the total number of cases and deaths in the general population (and unfortunately in occupational settings) were particularly significant.
The paper is not particularly innovative, as honestly acknowledged by Authors themselves, but it is well documented, deserving a potential publication that cannot be endorsed in its current stage of development - at least from the point of view of the present reviewer. However, I'm also certain that a series of minor improvements could be performed in a limited amount of time allowing a better appraisal and the eventual acceptance of the present paper. More precisely:
1) please provide as a first subheading of your methods section the settings of your study: main characteristics of your hospital (number of workers, number of total inpatients, main demographic characteristics of the total workforce: number of physicians, of nurses, etc, mean age...). Some information is provided in later sections, but a more detailed subheading would be better;
2) in order to better appraise the data reported in Table 1 and 3, data on Ratio could and should be improved by providing 95%CI; by assessment of 95%, the reader could appreciate whether for example 13.3% of ratio of 8th wave in doctor could be acknowledged as truly lower than 20.8% in nurse OR because of the overlap between confidence intervals a more precautionary approach should be encouraged.
3) In rows 179-184, Authors hint at the availability of some personal data about vaccination status of healthcare workers recruited from the parent center, but such information is not provided across the main text. Authors could improve the overall quality of their paper by providing this data in a specifically designed table.
Comments on the Quality of English LanguageSome minor typos are scattered across the main text; they could fix these issue by double checking the text before the resubmission.
Author Response
Dear Reviewer 4
We would like to thank you for carefully reading our paper and for your valuable advice.
I corrected the points you commented out as follows:
Revised parts are highlighted in blue in the manuscript for easy identification.
Reviewer 4:
Comments and Suggestions for Authors
Estimated Authors,
I've read with great interest your paper reporting on a single center experience during the later waves of COVID-19 pandemic in Japan. The present paper stresses how later waves, associated with Omicron VOC, were less frequently acknowledged for their morbidity and mortality, even though the total number of cases and deaths in the general population (and unfortunately in occupational settings) were particularly significant.
The paper is not particularly innovative, as honestly acknowledged by Authors themselves, but it is well documented, deserving a potential publication that cannot be endorsed in its current stage of development - at least from the point of view of the present reviewer. However, I'm also certain that a series of minor improvements could be performed in a limited amount of time allowing a better appraisal and the eventual acceptance of the present paper. More precisely:
- please provide as a first subheading of your methods section the settings of your study: main characteristics of your hospital (number of workers, number of total inpatients, main demographic characteristics of the total workforce: number of physicians, of nurses, etc, mean age...). Some information is provided in later sections, but a more detailed subheading would be better;
We added the information of our hospital in the method section.(line 70-74)
- in order to better appraise the data reported in Table 1 and 3, data on Ratio could and should be improved by providing 95%CI; by assessment of 95%, the reader could appreciate whether for example 13.3% of ratio of 8th wave in doctor could be acknowledged as truly lower than 20.8% in nurse OR because of the overlap between confidence intervals a more precautionary approach should be encouraged.
I have inserted the odds ratio and p-value in Table 3. (line 172-175)
3) In rows 179-184, Authors hint at the availability of some personal data about vaccination status of healthcare workers recruited from the parent center, but such information is not provided across the main text. Authors could improve the overall quality of their paper by providing this data in a specifically designed table.
The number of vaccinations must be recorded in the infectious disease reporting system. As for the content, I have omitted it due to space limitations, and I do not consider it to be important data to be presented in the table.
I attached the data in this response letter below for the reference.
(see the attached file)
Comments on the Quality of English Language
Some minor typos are scattered across the main text; they could fix these issue by double checking the text before the resubmission.
I am posting a version that has been proofread by an English proofreading expert. I mention this in my acknowledgments. (line 326)
We appreciate your review of our responses and revised versions. We sincerely hope that the revised version is satisfactory to the reviewers and deserves to be accepted.
Sincerely,
Authors

Round 2
Reviewer 4 Report
Comments and Suggestions for Authors
The paper has been improved according to my previous recommendations.
Therefore, I'm endorsing its acceptance.